# CLARREO Pathfinder/VIIRS Intercalibration: Quantifying the Polarization Effects on Reflectance and the Intercalibration Uncertainty

**Daniel Goldin[1,2,*], Xiaoxiong Xiong[3], Yolanda Shea[2] and Constantine Lukashin[2]**

1   Science Systems and Applications, Inc., (SSAI), Hampton, VA 23666, USA
2   NASA Langley Research Center, Hampton, VA 23666, USA
3   NASA Goddard Space Flight Center, Greenbelt, MD 20771, USA
*   Correspondence: daniel.goldin@nasa.gov

**Abstract:** Atmospheric scattering and surface polarization affect radiance measurements of polarization-sensitive instruments on orbit. Neglecting the polarization effects may lead to an inaccurate radiance/reflectance determination and underestimated radiance/reflectance uncertainty. Of the two instruments, CERES and VIIRS, slated to be intercalibrated by the CLARREO Pathfinder (CPF), the latter is known to be sensitive to polarization. The Pathfinder mission is tasked with accurately determining the uncertainty contribution of polarization and will provide the benchmark for the determination of the polarization correction factor for polarization-sensitive instruments. In this article, we show the formalism necessary to correct the reflectance for sensitivity to polarization after the CLARREO Pathfinder/VIIRS intercalibration, as well as the associated polarization uncertainty contribution to the overall intercalibrated reflectance error. To illustrate its usage, the formalism is applied to three dominant scene types.

**Keywords:** VIIRS; CLARREO Pathfinder; CPF; intercalibration; polarization; reflectance; reflectance correction; polarization uncertainty

## 1. Introduction

*CLARREO Pathfinder and VIIRS Missions*

In 2007, the National Research Council's Earth Science Decadal Survey recommended the implementation of Climate Absolute Radiance and Refractivity Observatory (CLARREO) [1] as a Tier-1 mission. The mission's objectives included making shortwave and infrared spectral radiance measurements at the unprecedented accuracy and serving as a calibration reference standard for other Earth-viewing instruments on orbit. Using CLARREO measurements, continuous climate data records may be constructed, which would lead to improved inputs to and assessment of climate models, thereby helping to inform sound policy decisions. In the early 2020s a CLARREO Pathfinder (CPF) mission [2] is slated to be launched and mounted on the International Space Station, with the goal of demonstrating the key measurement technologies needed by a full climate-observing mission, such as CLARREO. Specifically, it aims to demonstrate its ability to (a) make high accuracy SI-traceable reflectance measurements at 0.3% uncertainty through its on-orbit calibration and (b) transfer that calibration uncertainty to other sensors by intercalibrating CERES [3] and VIIRS [4] instruments at 0.3% uncertainty (the precision quoted here is at the $k = 1$ or $1\sigma$ level, which for a Gaussian distribution would correspond to a 68% confidence level). The CPF instrument is designed to be a Reflected Solar (RS) spectrometer operating in the 350–2300 nm spectral range. The cross-track at-nadir width is projected to be 70 km, with a resolution of 0.5 km.

The Visible/Infrared Imaging Radiometer Suite (VIIRS), which the CLARREO Pathfinder will intercalibrate, is a follow-on detector to the Moderate Resolution Imaging Spectroradiometer (MODIS) instruments currently taking data onboard the EOS Terra and Aqua satellites. The VIIRS instrument suite is mounted onboard the Suomi-NPP and NOAA-20 satellites launched in October 2011 and November 2017, respectively. Its calibrated and geolocated reflectance and radiance products are used to produce more than 20 Environmental Data Records (EDRs). The instrument is a whisk broom scanning radiometer consisting of the Rotating Telescope Assembly (RTA) and a double-sided Half-Angle Mirror (HAM), with the latter rotating at half the speed of the former [5]. Out of the total of 22 VIIRS bands, 14 Reflective Solar Bands (RSBs) cover the reflected solar region, with central wavelengths from 0.41–2.25 μm. The reflectance calibration uncertainty for those bands is about 2%, while the polarization sensitivity measured on the ground ranges from 2.5%–3% [6]. The cross-track width at nadir is 3060 km, with the resolution at nadir at 0.375 km for the three high-resolution RSB bands and 0.75 km for 11 moderate-resolution bands.

## 2. Materials and Methods

### 2.1. Intercalibration Procedure

During the intercalibration event, ISS, with the CLARREO Pathfinder instrument onboard, will fly below the orbit of a target imager, within the time window of less than a few (typically, less than five) minutes of the latter, sub-sampling the swath traversed by it. The CPF's gimbal design [7] will allow it to match the target imager's Viewing Zenith (VZA) and Relative Azimuth Angles (RAZ) (see Figure 1). The reflectance (or, equivalently, radiance) parametrization of the target imager, such as VIIRS, may then be expressed in terms of the Pathfinder's calibrated values. Typically, such parametrizations are linear, of the form: [8]

$$\rho_0 = A_0 + G_0 \rho_r, \tag{1}$$

where $\rho_r$ is the calibrated reflectance measured by the reference spectrometer (CPF), $A_0$ and $G_0$ are linear fit parameters, and $\rho_0$ is the reflectance measured by the target imager, such as VIIRS, after its intercalibration with the reference instrument. (The reflectance $\rho_r$ obtained by the reference imager itself can be calibrated using the same parametrization in Equation (1). In fact, such self-calibration will be performed by the Pathfinder using well-measured targets, such as the Sun and the Moon.) These parameters may be obtained from intercalibration over negligibly-polarized scenes, such as optically-thick clouds or snow.

Over polarized scenes, the reflectance needs to be corrected for polarization and the uncertainty due to this correction accounted for. Assuming the reference spectrometer itself is polarization insensitive or, alternatively, that its polarization has already been taken into account, the reflectance of the target spectrometer after intercalibration over polarized scenes may be expressed as:

$$\rho' = c\rho'_0 = c(A_0 + G_0\rho'_r), \tag{2}$$

where $\rho'$ and $\rho'_0$ are the corrected and uncorrected reflectances measured by the target imager, respectively, and $\rho'_r$ is the reflectance measured by the reference spectrometer (assumed to be calibrated) over polarized scenes. The coefficients $A_0$ and $G_0$ are assumed to be known from the prior unpolarized intercalibration (Equation (1)). The correction factor $c$ depends on the degree and angle of polarization, which vary by scene type, while its constant parameters are the diattenuation coefficient and the optical phase angle (see Sections 2.3 and 2.4 and Appendix A for further details). The essential part of the polarization correction during intercalibration is determining the values of the diattenuation coefficients and optical phases, allowing the target imager to measure correctly the reflectance at all times. The discussion of the correction factor $c$ and the contribution of the uncertainty contribution to the overall intercalibration error associated with it are the focus of this article.

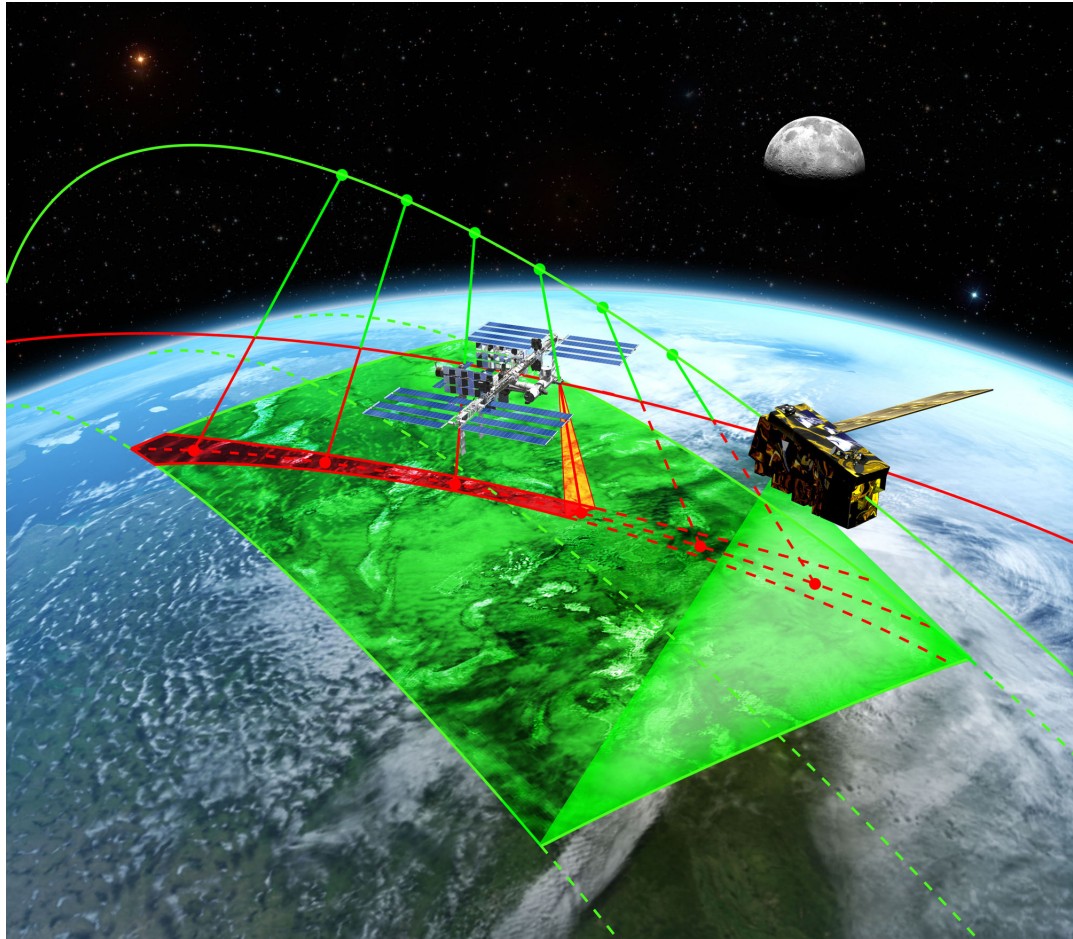

**Figure 1.** Rendering of the intercalibration of the CLARREO Pathfinder onboard the International Space Station (ISS) with the target imager, such as VIIRS.

## 2.2. Polarization and Polarization Distribution Models

The polarization of light may be described by four Stokes parameters, frequently denoted as $I$, $Q$, $U$, and $V$, where $I$ corresponds to the total intensity, and $Q$ and $U$ describe linear and $V$ circular polarizations. Since the circular polarization of the top-of-atmosphere reflected solar radiation has been found to be sufficiently small, $V$ is usually neglected [9]. The degree of linear polarization (polarization strength), denoted here by $P$ or $DOP$, can, therefore, be expressed in terms of the remaining three Stokes vectors as [10]:

$$P \equiv \frac{I_p}{I} = \frac{\sqrt{Q^2 + U^2}}{I},\tag{3}$$

while the angle of linear polarization (polarization orientation) $\chi$ is given by:

$$\chi = \frac{1}{2}\arctan(U/Q).\tag{4}$$

Thus, the polarization state of the reflected solar radiation is fully specified by the total intensity $I$, degree of linear polarization $P$, and angle of linear polarization $\chi$. Both $P$ and $\chi$ are sensitive to the viewing geometry, i.e., Relative Azimuth (RAZ), Viewing Zenith Angle (VZA), and the Solar Zenith Angle (SZA), as well as the scene type parameters, such as surface type, wind speed, cloud and aerosol optical depths, etc. In this work, we adopt the convention of the $P$ range varying between zero and one and $\chi$, between $0°$ and $180°$.

Since the CLARREO Pathfinder instrument is not designed to measure polarization parameters, in order to meet the stringent intercalibration uncertainty requirements, one must use polarization

lookup tables. These tables are derived from the Polarization Distribution Models (PDMs). Following the scheme proposed in [8], an individual PDM is simply a two-dimensional distribution of $P$ or $\chi$ in RAZ and VZA, with the rest of the parameters either fixed or constrained. Two types of PDMs will be used by the Pathfinder mission: the empirical and theoretical. The former are based on the polarization measurements obtained by the POLDER instrument [11], while the latter on the results from the Adding-Doubling Radiative Transfer Model (ADRTM) [12]. The POLDER instrument was mounted onboard the PARASOL satellite, active between 2004 and 2013. POLDER's Charge-Coupled Device (CCD) detector took measurements from nine spectral channels from blue (443 nm) to infrared (1020 nm), three of which—490, 670, and 865 nm—were polarized. The results discussed in this article rely on the Stokes parameters measured by the 865-nm channel. The empirical PDMs shown here represent global averages and standard deviations (used to estimate the contribution of polarization to the uncertainty in reflectance) of $P$ and $\chi$ as a function of RAZ and VZA, using the 2006 polarization data obtained by POLDER. For further details on the PDM construction, see [8,13]).

While both, the empirical and the theoretical PDMs, will cover the same range as the POLDER data, i.e., from 490–865 nm, the theoretical PDMs will be used to cover the regions of the visible spectrum beyond the reach of the POLDER instrument, as well as, possibly, the scenes with noisy or insufficient data. We note that there are multiple ways of combining empirical and theoretical PDMs to yield a final polarization value. The outline of one of the proposed methods is as follows. For given RAZ and VZA values, if the reflectance wavelength band falls within the range of the available POLDER bands, the mean of the theoretical and empirical values is taken. A confidence flag is assigned based on the level of agreement between the two values and the magnitude of empirical uncertainty. For a calibrated wavelength outside the POLDER range, a theoretical PDM value is taken. The exact functionality of the polarization algorithm is to be determined at a later date. We note that for the CPF wavelengths not within the POLDER bandwidths, the polarization variables $P$ and $\chi$ are linearly interpolated. Propagating the standard deviations in $P$ and $\chi$ using the interpolating function yield their respective uncertainties, $\sigma_P$ and $\sigma_\chi$. The multilinear interpolation function and its corresponding uncertainty has already been implemented for PDMs and will be described in a future publication.

### 2.3. Polarization Correction and Uncertainty for a Single-Imager Reflectance Measurement

As mentioned in Section 2.1, in the presence of a polarized scene, the radiance $\rho_0$ needs to be corrected for polarization effects. The relationship between the corrected radiance $\rho$ and the uncorrected one $\rho_0$ is simply:

$$\rho = c\rho_0, \tag{5}$$

where the correction factor $c$ depends on the degree of polarization $P$ and the angle of linear polarization $\chi$ as:

$$c = \frac{1}{1 + aP\cos 2(\chi + \phi)}, \tag{6}$$

with $a$ being the diattenuation coefficient and $\phi$ the phase angle (see Appendix A for the derivation). The diattenuation coefficient $a$ describes the instrument's polarization sensitivity, while the phase angle $\phi$ can vary with the band and detector, as well as the scan angle properties [6]. We note that the correction factor shown in Equation (6) is inline with that found in [6,14] and is more accurate than the one given in [8,13], where $\chi$ and $\phi$ dependence was neglected.

Denoting the uncertainties in $\rho_0$, $a$, $P$, and $\chi$ as $\sigma_a$, $\sigma_{\rho_0}$, $\sigma_P$, and $\sigma_\chi$, respectively, we apply the error propagation methods (for the derivation, see Appendix B) to estimate the contribution of polarization to the imager's reflectance uncertainty:

$$\sigma_\rho^2 = \left(\frac{\rho_0}{1 + aP\cos\theta}\right)^2 \left\{ \left(\frac{\sigma_{\rho_0}}{\rho_0}\right)^2 + \left(\frac{aP\cos\theta}{1 + aP\cos\theta}\right)^2 \left[\left(\frac{\sigma_a}{a}\right)^2 + \left(\frac{\sigma_P}{P}\right)^2 + 4\tan^2\theta(\sigma_\chi^2 + \sigma_\phi^2)\right] \right\} \tag{7}$$

with:

$$\theta \equiv 2(\chi + \phi). \tag{8}$$

Equation (7) may also be rewritten in terms of relative errors of the form $\sigma_x / \overline{x}$, where the subscript $x$ stands for either $\rho$, $\rho_0$, or $P$:

$$\delta_\rho^2 \equiv \left( \frac{\sigma_\rho}{\overline{\rho}} \right)^2 = \delta_{\rho_0}^2 + \left( \frac{aP \cos \theta}{1 + aP \cos \theta} \right)^2 \left\{ \delta_a^2 + \delta_P^2 + 4 \tan^2 \theta (\sigma_\chi^2 + \sigma_\phi^2) \right\}. \tag{9}$$

In the idealized case of the diattenuation coefficient exactly equal to zero in Equations (6) and (9), the instrument has no polarization sensitivity.

The formalism presented in this section is general enough that it can be applied to any single imager. In particular, it can be used for the case when the reference spectrometer is found to have a non-negligible polarization sensitivity and where an estimate of the reference imager's own contribution to the overall intercalibration uncertainty (described in the next section) is needed.

### 2.4. Polarization Contribution to the Overall Uncertainty in Reflectance after the Reference Detector/Target Imager Intercalibration

If the reference instrument is polarization insensitive, or, equivalently, has a negligible polarization sensitivity, as is the case of the CLARREO Pathfinder's design goal, the reflectance measured by it is approximately equal to the true Earth reflectance, i.e., $\rho \approx \rho_0$ (Equation (5)). In this case, only the target detector, such as VIIRS, needs to be intercalibrated using Equations (5) and (6). However, if the reference detector's optics are found to be polarization sensitive, the reflectance measured by it needs to be corrected first, making use of Equations (5) and (6). (We note that the CPF final polarization product would only provide CPF's own uncertainty due to polarization, and it would fall upon the target instrument's group to estimate the combined polarization uncertainty contribution, as shown in this section.) After this correction, the intercalibrated target imager's reflectance measurement will be accurate and unaffected by the reference detector's own polarization sensitivity; however, as a result, the overall reflectance uncertainty will increase. This is evident by examining the expression of uncertainty below and will be further demonstrated with a few examples in the next section (derivation in Appendix B):

$$\delta_{\rho'}^2 = \delta_{f_t}^2 + \delta_{f_{rt}}^2, \tag{10}$$

where:

$$\delta_{f_t}^2 = \delta_{A_0}^2 + \left( \frac{a_t P \cos \theta}{1 + a_t P \cos \theta} \right)^2 \left\{ \delta_{a_t}^2 + \delta_P^2 + 4 \tan^2 \theta (\sigma_\chi^2 + \sigma_{\phi_t}^2) \right\}, \tag{11}$$

using the definition of $\theta$ in Equation (8), while:

$$\delta_{f_{rt}}^2 = \delta_{G_0}^2 + \delta_{\rho_r^0}^2 + \left( \frac{AP \cos \Theta}{1 + AP \cos \Theta} \right)^2 \left\{ \delta_A^2 + \delta_P^2 + 4 \tan^2 \Theta (\sigma_\chi^2 + \sigma_\Phi^2) \right\}, \tag{12}$$

with:

$$\Theta = 2(\chi + \Phi)$$

$$\Phi \equiv \frac{1}{2} \arctan \left( \frac{a_t \sin 2\phi_t + a_r \sin 2\phi_r}{a_t \cos 2\phi_t + a_r \cos 2\phi_r} \right)$$

$$A \equiv \left[ (a_t \cos 2\phi_t + a_r \cos 2\phi_r)^2 + (a_t \sin 2\phi_t + a_r \sin 2\phi_r)^2 \right]^{1/2}.$$

As before, the relative uncertainties of the form $\sigma_x / \overline{x}$ are denoted by $\delta_x$. The subscripts $r$ and $t$ refer to the reference and target detectors, respectively, so that $a_r$ and $\phi_r$ are the diattenuation coefficient and the phase angles for the reference spectrometer and $a_t$ and $\phi_t$ are their equivalents for the target detector. $\delta_{\rho_r^0}$ is the uncertainty in uncorrected reflectance measured by the reference spectrometer.

$\delta_{A_0}$ and $\delta_{G_0}$ in Equations (11) and (12) are the uncertainties in the fit parameters in Equation (2). The uncertainty in the "combined" diattenuation coefficient is:

$$\delta_A^2 = \frac{1}{4}\left[\left(1 + \frac{a_t^2 - a_r^2}{A^2}\right)^2 \delta_{a_t}^2 + \left(1 - \frac{a_t^2 - a_r^2}{A^2}\right)^2 \delta_{a_r}^2\right]$$

and the total phase uncertainty is:

$$\sigma_\Phi^2 = \frac{1}{A^2(1 + t_\Phi^2)}\left[a_t^2(\cos 2\phi_t + t_\Phi \sin 2\phi_t)^2 \sigma_{\phi_t}^2 + a_r^2(\cos 2\phi_r + t_\Phi \sin 2\phi_r)\sigma_{\phi_r}^2\right],$$

where:

$$t_\Phi \equiv \tan 2\Phi.$$

Due to the lack of real data from the CLARREO Pathfinder, in what follows, we make an assumption that the reference spectrometer perfectly tracks the target spectrometer, VIIRS, setting the fit intercept $A_0 = 0$ and fit slope $G_0 = 1$. With these assumptions, the simplified relative intercalibration uncertainty derived in Appendix B used for the examples that follow is:

$$\delta_{\rho'}^2 = \delta_{\rho_r^0}^2 + \left(\frac{AP\cos\Theta}{1 + AP\cos\Theta}\right)^2\left\{\delta_A^2 + \delta_P^2 + 4\tan^2\Theta(\sigma_\chi^2 + \sigma_\Phi^2)\right\} \tag{13}$$

Equation (13) is analogous to Equation (9) and, in fact, reduces to the latter if the reference detector is insensitive to polarization, i.e., when $a_r \approx 0$. (One curious consequence of Equation (13) is that if one combines $a_r = a_t$ with $|\phi_r - \phi_t| = 90°$, it leads to the diattenuation coefficient $A = 0$, thereby eliminating the need for the PDMs.) In the next section, we will show three examples of the reflectance correction coefficients corresponding to three scene types and the associated uncertainties in reflectance due to polarization.

## 3. Results

In this section, we will apply the formalism in Section 2 to the three dominant scene types, the clear-sky ocean and water clouds and ice clouds above the ocean. We will try to give a sense of how much these two quantities vary by taking the lowest and the highest measured values of the diattenuation coefficient for VIIRS and the lowest and highest projected limits of accuracy on these coefficients for CPF.

All the datasets used in this section were derived from Level-1 and Level-2 *global* POLDER data [15,16] for the entire year 2006. Similarly to [8,13], we assume that the relative uncertainty $\delta_{\rho_0}$ is due to three sources: CPF's own instrument uncertainty, intercalibration uncertainty, and the residual uncertainty, not accounted for by the previous two sources of uncertainties. In all three examples, we will take the "worst-case scenario" and assume both the CPF instrument and intercalibration uncertainties to be 0.3% (see also the Introduction). We will assume the residual error to be 0.1%, as was done in [8,13]. Combining these three uncertainties leads to $\delta_{\rho_0} = 0.44\%$. In all three examples, we will neglect the uncertainties in the diattenuation coefficients and the optical phases. While these values are important to quantify, their contributions are expected to be smaller than that of the uncertainties due to $P$ and $\chi$.

### 3.1. Clear-Sky Ocean

While the clear-sky ocean ocean scene yields one of the lowest obtainable reflectances, it is characterized, at the same time, by some of the highest polarization values among the surface types identified by the International Geosphere-Biosphere Programme (IGBP) [17]. In this example, the chosen VIIRS band was M7, corresponding to the central wavelength of 862 nm, lying within the bandwidth of 33.7 nm [18] of the corresponding POLDER band of 865 nm. The latter band was used

to construct the empirical $P$ and $\chi$ PDMs (Figure 2, top). In order to demonstrate the magnitude and the spread of values of the correction factor $c_t$, we took the lowest and the highest values of the VIIRS diattenuation coefficient measured in the lab settings for a given wavelength. The lowest diattenuation coefficient $a_t = 0.0002$ was found for Detector 4, with Half-Angle Mirror (HAM) Side A and at scan angle $-45°$, while the highest, $a_t = 0.0049$, for Detector 10, HAM Side B, scan angle $-55°$. (Several configurations of the detector, HAM, and scan angle yielded approximately equal lowest and highest diattenuation coefficients, but for simplicity, we picked only one high and low value). The corresponding values of the phase angle $\phi_t$ were measured to be $136°$ and $-31°$, respectively. Using these values and the PDMs plotted in Figure 2, to compute the correction factor $c$ (Equation (6)), we obtained the plots in Figure 3. We observe that the correction factor was, essentially, one for $a_t = 0.0002$, and for $a_t = 0.0049$, it varied between 0.999 and 1.003, depending on RAZ and VZA.

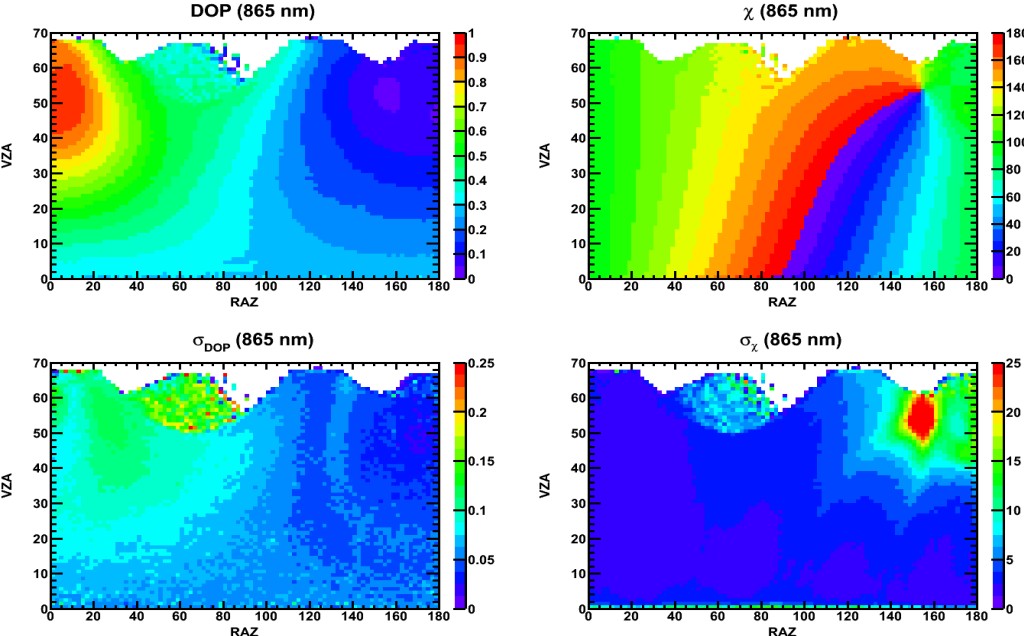

**Figure 2.** $P$ and $\chi$ PDMs for clear-sky ocean, subject to the selection criteria in Table 1. The figures in the bottom row show the corresponding absolute uncertainties in $P$ and $\chi$, which are used in estimating the contribution of polarization to the overall reflectance uncertainty.

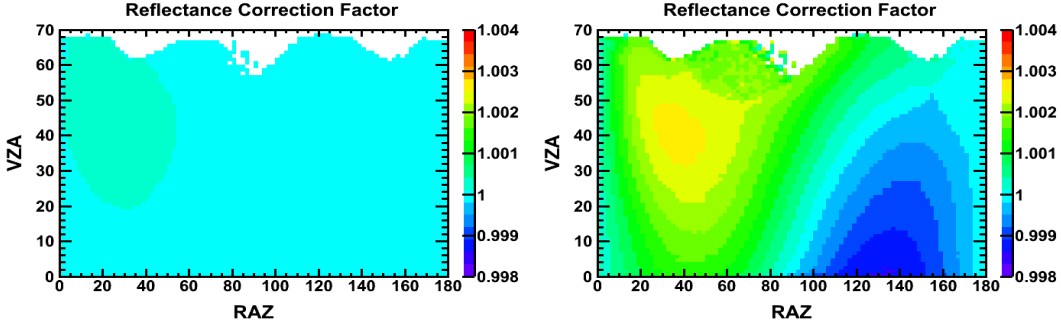

**Figure 3.** VIIRS Polarization correction factor $c$ for clear-sky ocean, using the PDMs shown in Figure 2. On the left, $c$ is plotted using $a_t = 0.0002$, $\phi_t = -31°$ (Band M7 (862 nm), HAM A, Detector 4, scan angle $= -45°$), while on the right with $a_t = 0.0049$, $\phi_t = 136°$ (Band M7 (862 nm), HAM B, Detector 10, scan angle $= -55°$).

Similar to the case of computing the correction coefficient, we would like to find the magnitude and the spread of values in RAZ and VZA for the polarization contribution to the uncertainty in reflectance $\delta_\rho$ (Equation (13)) for a given wavelength. To this end, we, once again, took the same lowest

and highest VIIRS values for $a_t$ as described in the previous paragraph and combined them with the lowest and highest targeted limits on the CPF diattenuation, $a_r = 0$ and $a_r = 0.005$, respectively. In the absence of the physical CPF instrument in both cases, we assumed $\phi_r = 0$ (our studies show that varying the phase angle does not change the average value of $c$ and $\delta_\rho$, but rather, shifted the maximum around the RAZ/VZA phase space). The resulting $\delta_\rho$ is plotted in Figure 4. The left-hand panel corresponds to the lower bound in $\delta_\rho$, i.e., $a_r = 0$, $a_t = 0.0002$, and $\phi_t = 136°$, while the right panel, to the upper bound, i.e., $a_r = 0.005$, $a_t = 0.0049$, and $\phi_t = -31°$. Propagating the error, as in Equation (13), we found that for $a_t = 0.0002$, the uncertainty in reflectance due to polarization was approximately constant at 0.44%, while for $a_t = 0.0049$, it varied between 0.44% and 4%, with the average uncertainty of 1.2%, as a function of RAZ and VZA.

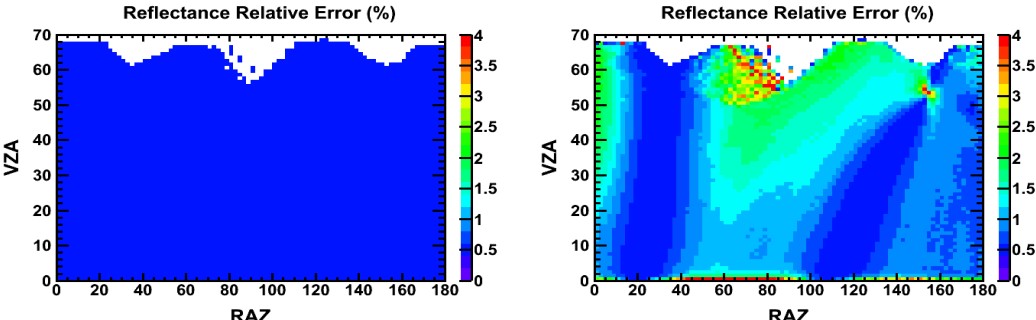

**Figure 4.** Polarization contribution to the overall relative reflectance uncertainty for clear-sky ocean after the CPF/VIIRS intercalibration. For the figure on the left, the CPF parameters $a_r = 0$ and $\phi_r = 0$ and the same VIIRS parameters as in the left panel of Figure 3 were used, while for the right-hand figure, $a_r = 0.005$ and $\phi_r = 0$, with the same VIIRS parameters as in the right panel of Figure 3, were taken.

Using the selection constraints on the POLDER data shown in Table 1, the empirical PDMs for $P$ and $\chi$ may be constructed. The results are shown in Figure 2.

**Table 1.** PDM constraints for selecting the clear-sky ocean scene.

| Constraint | Value/Range |
|---|---|
| Surface type: IGBP index [17] | 17 |
| POLDER band (nm) | 865 |
| SZA | $[50°, 60°]$ |
| Wind speed (m/s) | $[2., 10.]$ |
| Aerosol Optical Depth (AOD) | $[0.5, 1.]$ |

*3.2. Overcast Ocean: Water Clouds*

Next, we considered the case of the overcast ocean with thin water clouds. The PDM selection criteria are shown in Table 2.

**Table 2.** PDM constraints for selecting the overcast ocean with water clouds.

| Constraint | Value/Range |
|---|---|
| Surface type: IGBP index [17] | 17 |
| POLDER band (nm) | 865 |
| SZA (degrees) | $[30°, 40°]$ |
| Cloud Fraction | $[0.99, 1]$ |
| COT | $[1.0, 2.0]$ |
| Water cloud mask | 1 |

The empirical PDMs (in $P$ and $\chi$) corresponding to this scene type are shown Figure 5. We proceeded in the same way as described in the previous section. We used the same wavelength and VIIRS band, as well as the same values of $a_t$'s, $\phi_t$'s as in the previous section to obtain the lower and upper bounds on the correction factor $c$. The results are shown in Figure 6. From the figure, $c$ was close to unity for $a_t = 0.0002$ and seen to vary between 0.999 and 1.001, for $a_t = 0.0049$. The polarization contribution to the reflectance uncertainty with these values and the CPF diattenuation coefficient $a_r$ of zero and 0.005 (with $\phi_r = 0°$) were 0.44% at the lower bound and varied between 0.5% and 4% at the upper bound, with the average value at 1.0% (Figure 7).

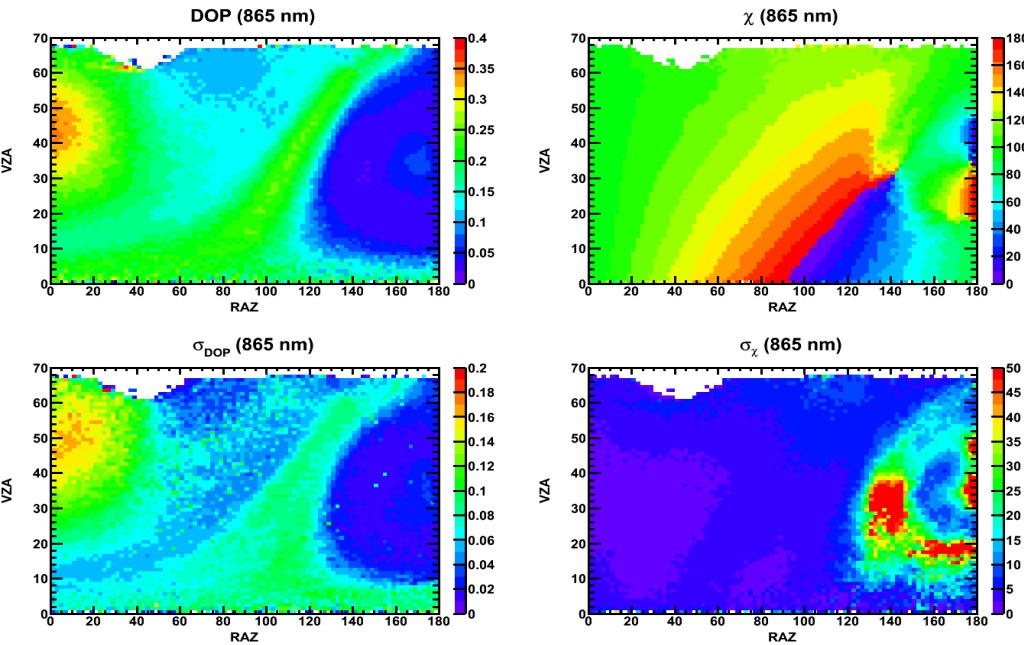

**Figure 5.** $P$ and $\chi$ PDMs for overcast ocean (water clouds), subject to the selection criteria in Table 3. The figures in the bottom row show the corresponding absolute uncertainties in $P$ and $\chi$, which are used in estimating the contribution of polarization to the overall reflectance uncertainty.

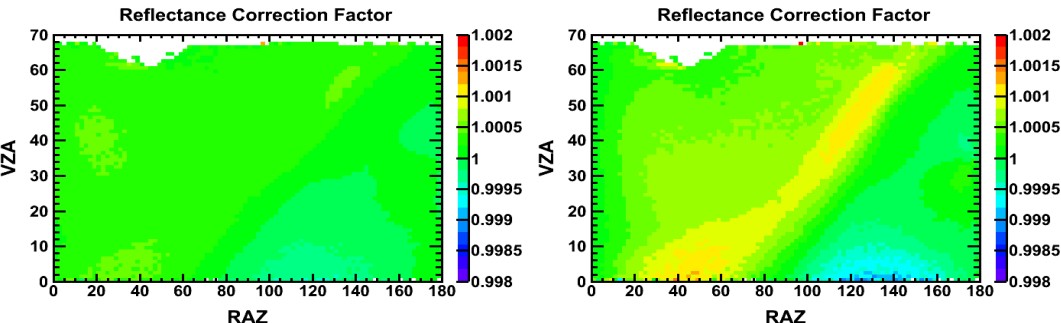

**Figure 6.** VIIRS polarization correction factor $c$ for overcast ocean (water clouds), using the PDMs shown in Figure 5. The left- and right-hand side figures are plotted using the same values as in the respective left- and right-hand side panels of Figure 3.

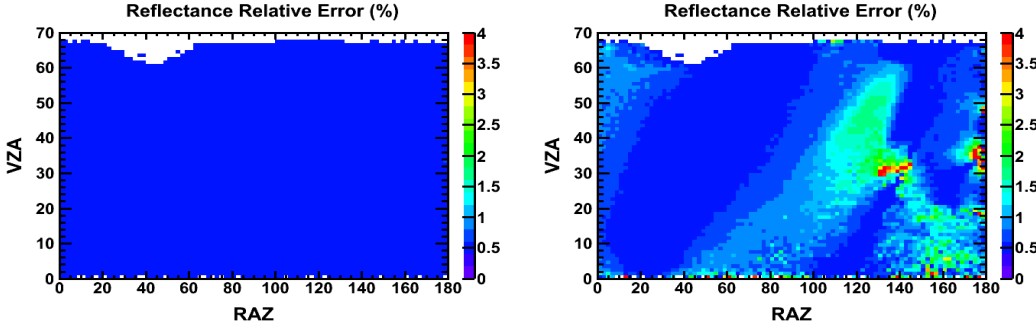

**Figure 7.** Polarization contribution to the overall relative reflectance uncertainty for overcast ocean (water clouds) after the CPF/VIIRS intercalibration. The left- and right-hand side figures are plotted using the same values as in the respective left- and right-hand side panels of Figure 4.

### 3.3. Overcast Ocean: Ice Clouds

Finally, we considered overcast ocean with ice clouds, with the selection criteria shown in Table 3.

**Table 3.** PDM constraints for selecting the overcast ocean with ice clouds.

| Constraint | Value/Range |
| --- | --- |
| Surface type: IGBP index [17] | 17 |
| POLDER band (nm) | 865 |
| SZA | $[30°, 40°]$ |
| Cloud fraction | $[0.99, 1]$ |
| COT | $[0.8, 1.2]$ |
| Ice cloud mask | 1 |

The empirical PDMs (in $P$ and $\chi$) corresponding to this scene types are shown Figure 8. The procedure and the input parameters to obtain the correction coefficient were the same as in the previous two examples. The correction factor $c$ was seen to vary between 0.998 and 1.005 for $a_t = 0.0002$ (Figure 9) and 0.996 and 1.006 for $a_t = 0.0049$. Finally, the contribution of polarization to the overall uncertainty in reflectance $\delta_\rho$ (Figure 10) was 0.44% for $a_t = 0.0002$ and $a_r = 0$ and was virtually constant at 0.54% for $a_t = 0.0049$ and $a_r = 0.005$, respectively.

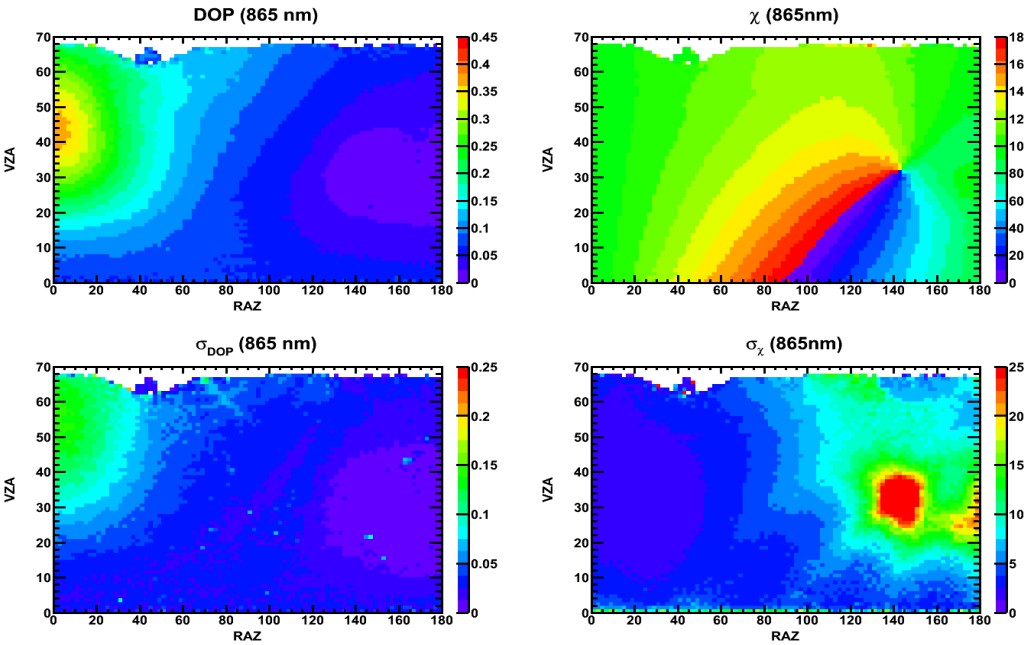

**Figure 8.** *P* and *χ* PDMs for overcast ocean (ice clouds), subject to the selection criteria in Table 3. The figures in the bottom row show the corresponding absolute uncertainties in *P* and *χ*, which are used in estimating the of polarization to the overall reflectance uncertainty.

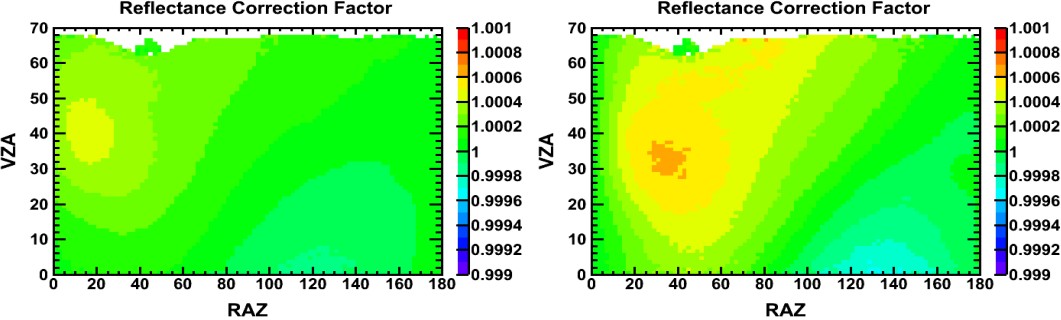

**Figure 9.** VIIRS Polarization correction factor *c* for overcast ocean (ice clouds), using the PDMs shown in Figure 8. The left- and right-hand side figures are plotted using the same values as in the respective left- and right-hand side panels of Figure 3.

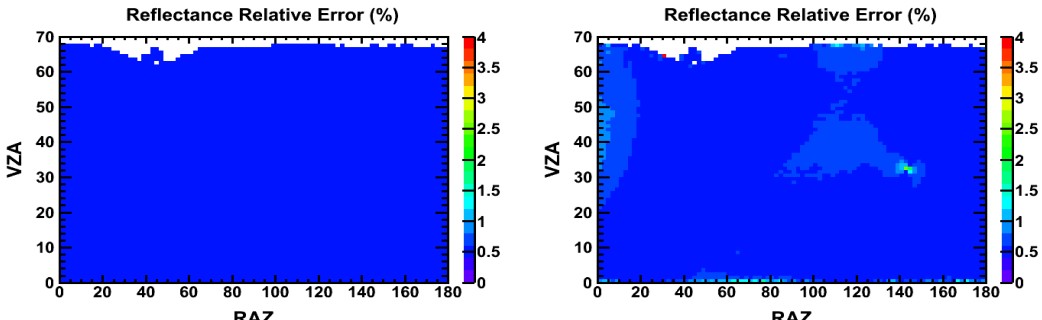

**Figure 10.** Polarization contribution to the overall relative reflectance uncertainty for overcast ocean (ice clouds) after the CPF/VIIRS intercalibration. The left- and right-hand side figures are plotted using the same values as in the respective left- and right-hand side panels of Figure 4.

## 4. Discussion

Although the three examples shown below do not represent an exhaustive list of scenes, however, they have been strategically chosen as the three limiting cases from which polarization for other scene types and geometries may be estimated. The ocean surface used in the first example has the highest polarizability among all the Earth surfaces, with any other surface yielding lower mean values of $P$. Likewise, in all the examples, the chosen solar zenith angle ranges encompassed Brewster's angle (Example 1) and cloudbows (Examples 2 and 3), corresponding to the highest levels of polarization for each scene; other SZA values yielded lower mean polarizations. For the third type of cloud phase, the mixed clouds, $P$ values were lower than water and higher than ice clouds, taken at similar optical depths. Similarly, for broken clouds, the polarization values lied between those for overcast and clear scenes, which are represented here. As for the wavelength coverage, it has been shown (see [13]) that the clear-sky ocean polarization does not vary much between 490 and 865 nm, so the upper bound on the correction factor shown here for 865 nm was valid for this entire wavelength range covered by the empirical PDMs. Finally, in all three examples, we chose the lowest (close to zero) and the highest measured diattenuation coefficient for VIIRS. The combination of these arguments allowed us to set upper and lower bounds on the polarization correction factor among all the viewable Earth scene types. A low value of the diattenuation coefficient chosen here also allowed us to set the lower bound on the polarization contribution to the overall reflectance uncertainty. Since we did not study all possible scene types, especially those with high relative uncertainties on $P$ and $\chi$ PDMs, we cannot yet claim to have established the upper bound on the polarization uncertainty. However, the uncertainties due to polarization shown here are expected to be representative of the bulk of the scene types.

In all the examples above, we assumed 0.005 as the diattenuation coefficient for the CLARREO Pathfinder. For the VIIRS channel with the lowest ground-based measurement of the diattenuation coefficient, 0.0002, the polarization had virtually no influence on the reflectance (correction factor $\approx 1$). Due to the low values of diattenuation in both instruments, the contribution of polarization on the intercalibration uncertainty then was dominated by the constant relative uncertainty $\delta_{\rho_0} = 0.4\%$. Choosing the VIIRS channels with the highest diattenuation resulted in sensitivity of the correction factor to variation in VZA and SZA. It was seen to vary between 0.996 and 1.003. The polarization contribution to the overall intercalibration uncertainty was significantly higher than in the low-diattenuation case, varying between 0.5% and 4%.

## 5. Conclusions

In this article, we have described the formalism necessary to (a) calculate the reflectance correction coefficient needed to account for the polarization sensitivity of an imager and (b) derive the polarization contribution to the overall reflectance uncertainty. Taking, alternately, the lowest and highest polarization sensitivities for VIIRS, we used our developed empirical PDMs and applied them to show how the polarization correction factor $c$ varies with RAZ and VZA for three scene types. We found it to be contained within the limits of 0.996 and 1.003. Depending on the sensitivity of the CPF and VIIRS instruments and the scene type, the contribution of polarization to the intercalibrated reflectance uncertainty was shown to vary between 0.44% and 4% at the extreme values and between 0.5% and 1% on average. Since the lowest value of the VIIRS diattenuation coefficient and the fact that the clear sky ocean scene with the highest polarization were chosen, we obtained the upper and lower bounds on the polarization correction factor for all Earth scenes. Likewise, the chosen VIIRS channel and geometry helped set the lower bound on the contribution of polarization to the reflectance uncertainty $\delta_\rho$. The upper bound on this value is expected to be higher for scene types with higher levels of relative uncertainties of PDMs; nevertheless, we consider the example shown to be representative of the bulk of the Earth scene types.

**Author Contributions:** Conceptualization, C.L. and D.G.; methodology, D.G.; software, D.G.; validation, D.G. ; formal analysis, D.G.; data curation, D.G and X.X.; writing—original draft preparation, D.G.; writing—review and editing, D.G. and Y.S.; project administration, C.L. and Y.S.; funding acquisition, C.L. and Y.S.

**Funding:** The authors are supported by NASA CLARREO Pathfinder funding.

**Acknowledgments:** The authors would like to thank Paul Smith of the Laboratory for Atmospheric and Space Physics (LASP) for his valuable input on this article.

**Conflicts of Interest:** The authors declare no conflict of interest. The funders had no role in the design of the study; in the collection, analyses, or interpretation of data; in the writing of the manuscript; nor in the decision to publish the results.

## Abbreviations

The following abbreviations are used in this manuscript:

| | |
|---|---|
| ADRTM | Adding-Doubling Radiative Transfer Model |
| AOD | Aerosol Optical Depth |
| CERES | Clouds and the Earth's Radiant Energy System |
| CLARREO | Climate Absolute Radiance and Refractivity Observatory |
| COT | Cloud Optical Thickness |
| CPF | Climate Absolute Radiance and Refractivity Observatory (CLARREO) Pathfinder |
| HAM | Half-Angle Mirror |
| IGBP | International Geosphere-Biosphere Programme index |
| ISS | International Space Station |
| MODIS | Moderate Resolution Imaging Spectroradiometer |
| RAZ | Relative Azimuth |
| RSB | Reflective Solar Band |
| RTA | Rotating Telescope Assembly |
| PARASOL | Polarization and Anisotropy of Reflectances for Atmospheric Sciences coupled with Observations from a Lidar |
| PDM | Polarization Distribution Model |
| POLDER | Polarization and Directionality of the Earth's Reflectances |
| SZA | Solar Zenith Angle |
| SI | Sysème Internationale d'Unités |
| VIIRS | Visible/Infrared Imaging Radiometer Suite |
| VZA | Viewing Zenith Angle |

## Appendix A. Deriving the Polarization Correction Factor

We note that the results presented here were previously derived in [6,14]. However, we thought it worthwhile to provide an alternative derivation, in order to provide an independent verification of these results. While [6,14] used Jones matrices, in this derivation, we used Mueller and Stokes formalism.

Let the initial state of the completely polarized light wave be given by the Stokes vector:

$$S = \begin{pmatrix} I \\ Q \\ U \\ V \end{pmatrix}, \tag{A1}$$

while the final state is:

$$S' = \begin{pmatrix} I' \\ Q' \\ U' \\ V' \end{pmatrix}. \tag{A2}$$

The Mueller transformation for the light passing through a linear polarizer ([10], p.18) is:

$$M = \frac{1}{2} \begin{pmatrix} \lambda_x + \lambda_y & \lambda_x - \lambda_y & 0 & 0 \\ \lambda_x - \lambda_y & \lambda_x + \lambda_y & 0 & 0 \\ 0 & 0 & 2\sqrt{\lambda_x \lambda_y} & 0 \\ 0 & 0 & 0 & 2\sqrt{\lambda_x \lambda_y} \end{pmatrix}, \tag{A3}$$

where $\lambda_x$ and $\lambda_y$ are the amplitude attenuation coefficients, squared, along the x- and y-axes, respectively. The initial to final polarization state transformation is:

$$S' = M \cdot S. \tag{A4}$$

Substituting Equations (A1)–(A3) into (A4), for the first Stokes parameter, the intensity $I'$, we obtain:

$$I' = \frac{1}{2} \left[ (\lambda_x + \lambda_y) I + (\lambda_x - \lambda_y) Q \right] \tag{A5}$$

Written in terms of the electrical field components in the Cartesian coordinates, the first two Stokes parameters $I$ and $Q$ are:

$$\begin{aligned} I &= E_x E_x^* + E_y E_y^* \\ Q &= E_x E_x^* - E_y E_y^*. \end{aligned} \tag{A6}$$

The products of the E-field components, including the space-time propagators, are:

$$\begin{aligned} E_x E_x^* &= (E_{0x} e^{i\delta(x,t)}) \cdot (E_{0x} e^{-i\delta(x,t)}) = E_{0x}^2 \\ E_y E_y^* &= (E_{0y} e^{i\delta(y,t)}) \cdot (E_{0y} e^{-i\delta(y,t)}) = E_{0y}^2. \end{aligned} \tag{A7}$$

Assuming positive $z$ direction as the direction of propagation of light and using spherical coordinates, we can rewrite the equations above as:

$$\begin{aligned} E_x E_x^* &= (E_0 \sin \psi \cos \chi)^2 = (E_0 \cos \chi)^2 \\ E_y E_y^* &= (E_0 \sin \psi \sin \chi)^2 = (E_0 \sin \chi)^2 \end{aligned} \tag{A8}$$

where $\psi$ is the E-field orientation angle (90°) and $\chi$ is the angle of linear polarization, which is referenced throughout this report. Thus, the first two Stokes parameters for the incident light (Equation (A6)) can be rewritten in terms of the E-field components (Equation (A8)) as:

$$\begin{aligned} I &= E_0^2 \\ Q &= E_0^2 \cos 2\chi = I \cos 2\chi \end{aligned} \tag{A9}$$

Substituting this into Equation (A5) gives:

$$I' = \frac{\lambda_x + \lambda_y}{2} \left[ 1 + \frac{\lambda_x - \lambda_y}{\lambda_x + \lambda_y} \cos 2\chi \right] I \tag{A10}$$

The expression above then relates the initial and final state intensity of completely polarized light passing though a linear polarizer. The partially-polarized light can be represented as a superposition of fully-unpolarized and completely-polarized states. Therefore, for the light incident on the detector, we may write:

$$S = (1 - P) \begin{pmatrix} I \\ 0 \\ 0 \\ 0 \end{pmatrix} + P \begin{pmatrix} I \\ Q \\ U \\ V \end{pmatrix}$$

$$= (1 - P) \begin{pmatrix} I \\ 0 \\ 0 \\ 0 \end{pmatrix} + P \begin{pmatrix} I \\ I\cos 2\chi \\ U \\ V \end{pmatrix}, \tag{A11}$$

where we made use of Equation (A9). Using the transformation associated with the polarizer (Equation (A4)), we obtain:

$$I' = \frac{\lambda_x + \lambda_y}{2} \left[ 1 + \frac{\lambda_x - \lambda_y}{\lambda_x + \lambda_y} P \cos 2\chi \right] I. \tag{A12}$$

In terms of the physical quantities, $\frac{\lambda_x + \lambda_y}{2}$ represents the transmittance of the polarizer, while the quantity $\frac{\lambda_x - \lambda_y}{\lambda_x + \lambda_y}$ is the diattenuation coefficient (denoted by $a_t$, $a_r$, or $A$ throughout the report). We note that for the idealized case of the polarization-insensitive detector, $\lambda_x = \lambda_y$, leading to the diattenuation coefficient of zero. We denote transmittance by $h$ and the diattenuation coefficient by $a$, rewriting Equation (A12) as:

$$I' = h \left[ 1 + aP \cos 2\chi \right] I \tag{A13}$$

Effects, such as the non-zero scan angle, optical retardation, etc., result in phase shifts of the angle of linear polarization, distorting the E-field components (see Equation (A8)):

$$E_x = E_0 \cos \chi \longrightarrow E_0 \cos(\chi + \phi)$$
$$E_y = E_0 \sin \chi \longrightarrow E_0 \sin(\chi + \phi)$$

so that the detected intensity, or radiance, in our case, given by Equation (A13), becomes:

$$I' = h \left[ 1 + aP \cos 2(\chi + \phi) \right] I. \tag{A14}$$

For a detector measurement that takes into account the optical transmittance, but not the polarization, the at-detector radiance $I_{det}$ and incident radiance $I_{inc}$ are thus related as:

$$I_{det} = \left[ 1 + aP \cos 2(\chi + \phi) \right] I_{inc}, \tag{A15}$$

implying that in order to obtain the correct incident radiance, the at-detector radiance needs to be corrected by a factor of $1/[1 + aP \cos 2(\chi + \phi)]$. Since for a given solar zenith angle, the radiance is proportional to reflectance $\rho$, for the corrected radiance incident on a detector, we may write:

$$\rho = \frac{\rho_0}{1 + aP \cos 2(\chi + \phi)}, \tag{A16}$$

where $\rho_0$ is the uncorrected radiance. This is the formula given in Equations (5) and (6).

### Appendix B. Deriving the Reflectance Uncertainty due to Polarization

In order to derive the uncertainty in reflectance due to polarization in the most general way, we assumed that both the target and the reference intercalibrating instrument are polarization sensitive, i.e., $a_t \neq 0$ and $a_r \neq 0$. The corrected reflectance after the intercalibration between the reference and target detector is given by Equation (2):

$$\rho' = c_t \rho_0' = c_t (A_0 + G_0 \rho_r'), \tag{A17}$$

where $c_t$ is the polarization correction factor for the target spectrometer and $A_0$ and $G_0$ are the fit parameters described in Section 2.1. However, if the reference spectrometer itself needs to be corrected for polarization, its own measured reflectance is given by:

$$\rho_r' = c_r \rho_r^0, \tag{A18}$$

where $c_r$ is the polarization correction factor for the reference spectrometer and $\rho_r^0$ is its measured uncorrected reflectance. Substituting Equation (A18) into Equation (A17), we obtain:

$$\rho' = c_t \rho_0' = c_t (A_0 + G_0 c_r \rho_r^0). \tag{A19}$$

Using Equation (6), this can be rewritten as:

$$\rho' = \frac{A_0}{1 + a_t P \cos 2(\chi + \phi_t)} + \frac{G_0 \rho_r^0}{(1 + a_t P \cos 2(\chi + \phi_t))(1 + a_r P \cos 2(\chi + \phi_r))} \tag{A20}$$

This is the most general formula describing intercalibration between the reference and target detector, including the correction for polarization. In a more compact Equation (A20):

$$\rho' \equiv f_t(P, \chi) + f_{rt}(P, \chi), \tag{A21}$$

with:

$$f_t(P, \chi) \equiv \frac{A_0}{1 + a_t P \cos 2(\chi + \phi_t)} \tag{A22}$$

and:

$$f_{rt}(P, \chi) \equiv \frac{G_0 \rho_r^0}{(1 + a_t P \cos 2(\chi + \phi_t))(1 + a_r P \cos 2(\chi + \phi_r))} \tag{A23}$$

Thus, the overall relative intercalibration uncertainty may be represented as:

$$\delta_{\rho'}^2 = \delta_{f_t}^2 + \delta_{f_{rt}}^2, \tag{A24}$$

written in terms of relative errors of the form $\sigma_x / \bar{x}$. We first start with the second term. Since the diattenuation coefficients are typically of the $10^{-3}$–$10^{-2}$ order of magnitude, the term containing their product $a_t a_r$ in Equation (A23) may be neglected to yield:

$$\rho \approx \frac{G_0 \rho_r^0}{1 + AP \cos 2(\chi + \Phi)}, \tag{A25}$$

where $A$ and $\Phi$ is a combined reference/target imager correction factor, diattenuation coefficient and a phase angle, respectively, with:

$$A \equiv \left[ (a_t \cos 2\phi_t + a_r \cos 2\phi_r)^2 + (a_t \sin 2\phi_t + a_r \sin 2\phi_r)^2 \right]^{1/2}$$
$$\Phi \equiv \frac{1}{2} \arctan \left( \frac{a_t \sin 2\phi_t + a_r \sin 2\phi_r}{a_t \cos 2\phi_t + a_r \cos 2\phi_r} \right). \tag{A26}$$

In the definition above, $a_t$ and $a_r$ are the diattenuation coefficients and $\phi_t$ and $\phi_r$ are the phase angles for the reference and target instruments, respectively.

Representing the formula above in a more compact form as:

$$\rho \equiv \frac{G_0 \rho_r^0}{f(P, \chi)}. \tag{A27}$$

and using error propagation on the formula above, the uncertainty in reflectance is determined by the uncertainties in $P$, $\chi$, $\rho_r^0$, $A$, and $\Phi$ denoted as $\sigma_P$, $\sigma_\chi$, $\sigma_{\rho_r^0}$, $\sigma_A$, and $\sigma_\Phi$, respectively:

$$
\sigma_\rho^2 = \left(\frac{\sigma_{G_0}}{f}\right)^2 + \left(\frac{\sigma_{\rho_r^0}}{f}\right)^2 + \left(\frac{\rho_r^0}{f^2}\frac{\partial f}{\partial A}\sigma_A\right)^2 + \left(\frac{\rho_r^0}{f^2}\frac{\partial f}{\partial \Phi}\sigma_\Phi\right)^2
$$
$$
+ \left(\frac{\rho_r^0}{f^2}\frac{\partial f}{\partial P}\sigma_P\right)^2 + \left(\frac{\rho_r^0}{f^2}\frac{\partial f}{\partial \chi}\sigma_\chi\right)^2 + 2\left(\frac{(\rho_r^0)^2}{f^4}\right)\frac{\partial f}{\partial P}\frac{\partial f}{\partial \chi}\sigma_{P\chi}, \tag{A28}
$$

where for the reasons of compactness, we have omitted the $P$ and $\chi$ dependence of $f$. As we have done for the derivation of Equation (A25), we can neglect the covariance term $\sigma_{P\chi}$ since it is quartic in the combined diattenuation coefficient $A$, so that:

$$
\sigma_\rho^2 \approx \left(\frac{\sigma_{G_0}}{f}\right)^2 + \left(\frac{\sigma_{\rho_r^0}}{f}\right)^2 + \left(\frac{\rho_r^0}{f^2}\frac{\partial f}{\partial A}\sigma_A\right)^2 + \left(\frac{\rho_r^0}{f^2}\frac{\partial f}{\partial \Phi}\sigma_\Phi\right)^2 + \left(\frac{\rho_r^0}{f^2}\frac{\partial f}{\partial P}\sigma_P\right)^2 + \left(\frac{\rho_r^0}{f^2}\frac{\partial f}{\partial \chi}\sigma_\chi\right)^2. \tag{A29}
$$

Written explicitly:

$$
\sigma_\rho^2 = \left(\frac{G_0\rho_r^0}{1+AP\cos 2(\chi+\Phi)}\right)^2
$$
$$
\times \left\{\left(\frac{\sigma_{G_0}}{f}\right)^2 + \left(\frac{\sigma_{\rho_r^0}}{\rho_r^0}\right)^2 + \left(\frac{AP\cos 2(\chi+\Phi)}{1+AP\cos 2(\chi+\Phi)}\right)^2\left[\left(\frac{\sigma_P}{P}\right)^2 + 4(\sigma_\chi^2+\sigma_\Phi^2)\tan^2 2(\chi+\Phi)\right]\right\}. \tag{A30}
$$

In terms of relative uncertainties, the second term in the overall relative intercalibration uncertainty (Equation (A24)):

$$
\delta_{f_{rt}}^2 = \delta_{G_0}^2 + \delta_{\rho_r^0}^2 + \left(\frac{AP\cos\Theta}{1+AP\cos\Theta}\right)^2\left\{\delta_A^2 + \delta_P^2 + 4\tan^2\Theta(\sigma_\chi^2+\sigma_\Phi^2)\right\}, \tag{A31}
$$

with $\delta_A$ and $\sigma_\Phi$ given in Section 2.4 and $\Theta$ defined as:

$$
\Theta = 2(\chi+\Phi). \tag{A32}
$$

We note that due to the angle being a circular quantity, the absolute error on the angle of linear polarization was used here.

The first term in Equation (A24) follows easily by comparing it with the second term above:

$$
\delta_{f_t}^2 = \delta_{A_0}^2 + \left(\frac{a_t P\cos\theta}{1+a_t P\cos\theta}\right)^2\left\{\delta_{a_t}^2 + \delta_P^2 + 4\tan^2\theta(\sigma_\chi^2+\sigma_{\phi_t}^2)\right\}, \tag{A33}
$$

with:

$$
\theta = 2(\chi+\phi_t). \tag{A34}
$$

If one is interested in estimating the uncertainty contribution of the single imager only, the result is obtained by simply setting the diattenuation coefficient of the second imager to zero in Equation (A26). Therefore, if the contribution to the intercalibration uncertainty of the CPF by itself is required, $a_t$ is set to zero, and Equation (A31) reduces to the form shown by Equation (9), i.e.,:

$$
\delta_\rho^2 \equiv \frac{\sigma_\rho^2}{\rho^2} = \delta_{\rho_r^0}^2 + \left(\frac{a_r P\cos 2(\chi+\phi_r)}{1+a_r P\cos 2(\chi+\phi_r)}\right)^2\left\{\delta_a^2 + \delta_P^2 + 4(\sigma_\chi^2+\sigma_{\phi_r}^2)\tan^2 2(\chi+\phi_r)\right\} \tag{A35}
$$

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
