# Peer review of "CLARREO Pathfinder/VIIRS Intercalibration: Quantifying the Polarization Effects on Reflectance and the Intercalibration Uncertainty"

_remotesensing, doi:10.3390/rs11161914_

Round 1

Reviewer 1 Report

Polarization sensitivity is an factor influencing high quality radiation data. This paper describes the method to correct the reflectance for sensitivity to polarization and derive the associated polarization uncertainty in intercalibration. Some suggestions are listed below.

1. In the abstract, it is said that CLARREO will provide the benchmark for the determination of the polarization correction factor. It is recommended to give more supporting information in the introduction about how to achieve this goal. 

2. To make it more understandable, it is recommended to add the description about the general intercalibration between CLARREO and target spectrometer to get the unbiased reflectance, accounting for polarization and other effects.

3. To correct for the polarization effect, the DOP and angle of linear polarization must be estimated by PDMs. What about the uncertaintie of PDMs derived from POLDER data or RTM simulations?  Could the PDMs of 865nm be applied to other wavelengths? 

4. Line 153, it seems that Eq.7 should be Eq. 8. 

5. According to the results, the contribution of polarization in intercalibration varies between 0.2% to 0.24% at 865 nm, what about the range at other wavelength?

6. Line 183-184, "even at the upper......CPF is insensitive to the polarization effects". It's somewhat confusing. It is recommended to make it clearer. 

7. In P13, Eq.(A17), rho_0 is the uncorrected reflectance measured by reference senor, according to Eq.(A16) and (3), rho_r=c_r*rho_0, is the correct incident reflectance of reference senor. What's the meaning of rho=c_t*rho_r? It's confusing. 

Author Response

Pdf attached.

Reviewer 2 Report

It is an interesting research paper, showing how to improve VIIRS reflectance measurement by correcting polarization. I suggest to accept it after a minor revision.

1. It is stated that the empirical PDMs represent global averages and standard deviations, which means the empirical PDMs are developed under very diversity of surface and atmospheric conditions, however, it is not clear whether the theoretical PDMs are consistent with the empirical PDMs, in other words, whether the theoretical PDMs are developed under consistent conditions as that of the empirical PDMs. Some more words on the theoretical PDMs should be added.

2. The results are presented under three conditions, i.e., clear-sky ocean, overcast ocean over water and ice clouds. Because it is not clear how to develop the theoretical PDMs, it is not easy to understand the logic. Additionally, because the empirical PDMs are developed based on global mean POLDER measurements, it seems not possible to have information about clear-sky and cloudy-sky from the empirical PDMs. 

3. The abbreviation should be provide with explanation in the first occurrence. 

Author Response

PDF file attached.

Reviewer 3 Report

There is almost nothing but positive comments to say about this manuscript. Polarization is often an overlooked complication of instrument performance, and the authors have analyzed its effects on the correction factor for radiance and associated uncertainties.  

This is analyzing the potential, upcoming CLARREO Pathfinder mission, which will be intercalibrated with various A-train instruments.  

With one minor correction that was detected, this paper appears otherwise ready for publication, and it should be helpful to the remote-sensing community.  

The correction is that the "amplitude attenuation coefficient" is called the "absorption coefficient" in Appendix A (e.g., lambda_x, lambda_y).  The point is twofold here:

(1.) one needs to specify field AMPLITUDE, not flux (the latter being related to the square of the field.

(2.) absorption coefficient is used as a different term elsewhere in the sciences; however, the book be E. Collett, "Polarized Light: Fundamentals and Applications," LOC QC441.C65 1992, uses the term "amplitude attenuation coefficient."  

Overall, this is unfortunate, because the coefficient is 1 when there is zero attenuation, but that is the nomenclature.  

If it suits the journal, the authors might consider adding a list of acronyms appendix as well.  

Author Response

PDF attached.
